# Superficial Inguinal Lymph Nodes for Screening Dead Pigs for African Swine Fever

**DOI:** 10.3390/v14010083

**Published:** 2022-01-04

**Authors:** Kalhari Bandara Goonewardene, Chukwunonso Onyilagha, Melissa Goolia, Van Phan Le, Sandra Blome, Aruna Ambagala

**Affiliations:** 1Canadian Food Inspection Agency, National Center for Foreign Animal Disease, Winnipeg, MB R3E 3M4, Canada; kalhari.goonewardene@inspection.gc.ca (K.B.G.); chuckwunonso.onyilagha@inspection.gc.ca (C.O.); melissa.goolia@inspection.gc.ca (M.G.); 2Department of Microbiology and Infectious Disease, College of Veterinary Medicine, Vietnam National University of Agriculture, Hanoi 100000, Vietnam; letranphan@vnua.edu.vn; 3Institute of Diagnostic Virology, Friedrich-Loeffler-Institut, 17493 Greifswald, Germany; Sandra.Blome@fli.de; 4Department of Comparative Biology, Faculty of Veterinary Medicine, University of Calgary, Calgary, AB T2N 1N4, Canada

**Keywords:** African swine fever, superficial inguinal lymph node, dead pig surveillance, screening, alternative samples

## Abstract

African swine fever (ASF) has spread across the globe and has reached closer to North America since being reported in the Dominican Republic and Haiti. As a result, surveillance measures have been heightened and the utility of alternative samples for herd-level monitoring and dead pig sampling have been investigated. Passive surveillance based on the investigation of dead pigs, both domestic and wild, plays a pivotal role in the early detection of an ASF incursion. The World Organization for Animal Health (OIE)-recommended samples for dead pigs are spleen, lymph nodes, bone marrow, lung, tonsil and kidney. However, obtaining these samples requires opening up the carcasses, which is time-consuming, requires skilled labour and often leads to contamination of the premises. As a result, we investigated the suitability of superficial inguinal lymph nodes (SILNs) for surveillance of dead animals. SILNs can be collected in minutes with no to minimum environmental contamination. Here, we demonstrate that the ASF virus (ASFV) genome copy numbers in SILNs highly correlate with those in the spleen and, by sampling SILN, we can detect all pigs that succumb to highly virulent and moderately virulent ASFV strains (100% sensitivity). ASFV was isolated from all positive SILN samples. Thus, sampling SILNs could be useful for routine surveillance of dead pigs on commercial and backyard farms, holding pens and dead on arrival at slaughter houses, as well as during massive die-offs of pigs due to unknown causes.

## 1. Introduction

African swine fever (ASF) is undoubtedly the most critical pig disease affecting the global swine industry at present [1]. It continues to spread in Europe, Asia and sub-Saharan Africa. The most recent ASF outbreaks in the Dominican Republic [2] and Haiti [3] have heightened concerns for the North American swine industry regarding preventing the entry of this deadly pig disease to the North American region [4]. ASF virus (ASFV), the causative agent of ASF, is a large, enveloped double-stranded DNA arbovirus of the family *Asfarviridae* [5]. One of the greatest challenges faced by ASF-free countries is detecting an ASFV incursion as quickly as possible to minimize the economic and social impact due to associated high mortality, stamping out, trade sanctions and regaining the disease-free status. For early detection of ASF, many countries have implemented surveillance programs for both domestic and wild pigs [6,7].

Passive surveillance based on testing dead pigs is one of the most effective ways to detect an ASF incursion [7,8,9]. Screening of dead pigs in large commercial farms, backyard farms, markets, or packing plants and those that are dead on arrival at the slaughterhouses as part of ongoing surveillance increases the probability of detecting an ASF incursion into disease-free areas. The current OIE-recommended sample types for ASF detection in dead animals include spleen, lymph nodes, bone marrow, lung, tonsil and kidney [10]. Harvesting the OIE-recommended sample matrices requires opening up the carcass and a full necropsy, which is time-consuming, requires highly skilled staff, instrumentation and additional resources for cleaning and disinfection of the contaminated areas after sample collection.

Highly virulent ASFV p72 genotype II strains that originated from ASFV Georgia 2007/1 [11] are responsible for the ongoing ASF outbreaks in Europe, Russia, Asia and in the Dominican Republic [2] and Haiti. They cause per-acute or acute disease in domestic and wild pigs and most of the infected pigs succumb to the disease within two weeks post-infection. As the ASF outbreak continues to spread through the domestic and wild boar population globally, a number of less virulent ASFV strains has emerged [12]. One of them, ASFV Estonia 2014, led to a higher number of clinically healthy, antibody-positive animals in northeastern Estonia and is considered a moderately virulent strain, especially for domestic pigs [13,14].

The main route of ASFV transmission in pigs is by oro-nasal route via direct contact with an infected animal or fomites, or feeding on contaminated swill or feed etc. [15]. Following oro-nasal entry, ASFV replicates in the tonsils and regional lymph nodes and then spreads through the lymph and blood to secondary lymphoid organs within 2–3 days [16,17,18]; this makes lymphoid organs an ideal tissue type for early detection of ASFV. Among the lymphoid organs, spleen is considered the best organ for ASFV detection in pigs, but access to it requires opening up the carcass [19]. In pigs, a handful of lymph nodes such as submandibular, superficial cervical, sub-iliac and superficial inguinal lymph nodes can be accessed through the skin without opening the carcasses [20]. Among the superficial lymph nodes, superficial inguinal lymph nodes (SILNs) are the largest and the most accessible through the skin. Harvesting SILNs requires no specialized skills and can be completed in less than a minute using a disposable scalpel and a pair of forceps (Figure 1) with minimum bleeding [21,22].

The objective of this study was to evaluate the ability to use this easily accessible lymphoid organ in pigs that succumbed to ASF infection as an alternative to opening the carcasses to obtain spleen and other OIE-recommended tissue samples.

## 2. Materials and Methods

### 2.1. Clinical Samples

The samples used in this study were collected from six different animal experiments conducted at the biosafety level 3 facility at the National Center for Foreign Animal Diseases (NCFAD), Winnipeg, MB, Canada. The use of animals for the experiments was approved by the Animal Care Committee at the Canadian Science Centre for Human and Animal Health (CSCHAH; AUD-C-19-008, AUD-C-19-012 and AUD-C-21-003). All procedures involving animals were performed in agreement with the Canadian Council for Animal Care guidelines. In all experiments, 4–5-week-old weaned Landrace × Duroc × Large white crossbred pigs obtained from a local supplier in Winnipeg were used. Once arrived, the pigs were assigned to designated pens and provided with a commercial ration twice a day and ad libitum water throughout the experiments. Following a seven-day acclimatization period, the pigs were inoculated with the corresponding viruses. Experiments 1, 3, 4 and 5 were performed with the purpose of understanding pathogenesis and collection of clinical samples using different ASFV field isolates.

In experiment 1, three pigs (Table 1, pigs 1–3) were infected oro-nasally with ASFV Estonia 2014 at a 2 × 10^5^ TCID_50_ dose in a total volume of 2 mL of virus/pig; 0.5 mL in each nostril and 1 mL orally. By 6 days post-infection (dpi), pigs started to show elevated rectal temperatures and lethargy. The three pigs reached humane endpoints at 9 (pig 2), 10 (pig 1) and 17 (pig 3) dpi.

In experiment 2 (Table 1, pigs 4–18), pigs were infected oro-nasally with ASFV Estonia 2014 at a dose of 2 × 10^5^ TCID_50_ per pig in 2 mL total volume. Pigs developed fever around 4 dpi and started becoming quiet and lethargic. At 7 dpi, the first pig (pig 18) was found dead. The first dead pig and every pig found dead or euthanized thereafter were considered for this analysis. Whole blood, SILN and spleen samples were collected from pigs that were euthanized as they reached humane endpoints and spleen and SILNs were collected from the ones that were found dead.

Experiment 3 was conducted by infecting two pigs (Table 1, pigs 19–20) with ASFV Ghana 20 (Genotype I, isolate from Oyibi, Ghana, in 2017) intramuscularly at 1 × 10^3^ TCID_50_ per pig in 2 mL in the hind thigh (1 mL per thigh). At 5 dpi, both pigs started developing fever, lethargy and the disease progressed very rapidly, resulting in hematochezia. By 7 dpi, one of the two pigs (pig 19) was found dead and the other was euthanized as it reached a humane endpoint.

In experiment 4, two pigs (Table 1, pigs 21 and 22) were infected oro-nasally with 3 mL of ASFV Nigeria RV502, an isolate from the 2020 ASF genotype II outbreak from Nigeria [23], at 2 × 10^5^ TCID_50_ per pig. Both pigs developed fever around 4 dpi with dried and dark fecal pellets and lethargy. By 5 dpi, both pigs developed rectal bleeding and at 6 dpi, one of the pigs (pig 21) was found dead and, on the same day, the other pig (pig 22) was humanely euthanized.

In the experiment 5, three pigs (Table 1, pigs 23–25) were infected with 2 × 10^5^ TCID_50_/pig of ASFV Vietnam 2561 (genotype II strain from Hung Yen province, Vietnam, in 2020) in a total volume of 2.8 mL oro-nasally. Starting from 5 dpi, the pigs developed fever. On 9 dpi, one of the pigs (pig 23) was moderately depressed and died during sampling under anesthesia. By 12 dpi, the remaining two pigs were found highly depressed and were euthanized as they reached humane endpoints.

Experiment 6 (Table 1, pigs 26–44) was conducted as one of four experiments to assess the feasibility of using swine oral fluids as an aggregate sample type to detect ASF genome in commercial pens [24]. In this experiment, a pig (seeder pig) (Table 1, pig 35) was randomly selected from the group and infected with a 1 × 10^3^ TCID_50_ dose of ASFV Georgia 2007/01, 1 mL, intramuscularly in the left thigh. Upon recovery from anesthesia, the seeder pig was sent back to the group to mingle with the rest of the contact pigs. The seeder pig developed clinical signs and was found dead at 7 dpi. The contact pigs started to develop clinical signs around 8 days post-exposure (dpe) to the seeder pig. Blood was collected from each pig every other day under general anesthesia with isoflurane 2 ppm/h, inhalation delivered in 100% oxygen. Samples from nineteen pigs from this experiment that died or were euthanized as they reached humane endpoints were used in this study.

In all experiments described above, whole blood samples were collected from the external jugular vein under general anesthesia with isoflurane 2 ppm/h, inhalation delivered in 100% oxygen either on the day of humane endpoint or the day before. From all pigs that were found dead and those euthanized, superficial inguinal lymph nodes were collected through a short incision in the inguinal region prior to opening up the carcasses. Spleen samples were collected after opening up the carcasses.

### 2.2. ASFV Genome Detection

Tissue homogenates (10% *w*/*v*) of the spleen and SILN tissues were prepared using Precellys lysing kits with the help of a Precellys tissue homogenizer (Bertin, France). Total nucleic acid extraction from whole blood and tissue homogenates was performed as described previously [24,25], using a MagMax™ Pathogen RNA/DNA kit (Thermofisher Scientific, Waltham, MA, USA), according to the low cell count protocol provided by the manufacturer on the MagMax™ Express-96 Deep Well Magnetic Particle Processor (Thermofisher Scientific, Waltham, MA, USA).

ASFV genomic DNA was detected using a quantitative real-time PCR (qRT-PCR) assay targeting a highly conserved region of the p72 open reading frame. The limit of detection of the real-time PCR assay is between 5.7 and 57 copies of the ASFV genome [26]. A qRT-PCR assay specific for β-actin (internal control) was used to control for efficient extraction and amplification [27]. The qRT-PCR reaction was prepared using TaqMan™ Fast Virus 1-Step Master Mix (Thermofisher Scientific, Waltham, MA, USA) and was amplified using the Bio-Rad CFX96 instrument (Bio-Rad, Mississauga, ON, Canada) using the recommended cycling conditions (50 °C for 5 min; 95 °C for 20 s, followed by 95 °C for 3 s and 60 °C for 30 s). The Ct values were converted to viral genomic log_10_ copy numbers based on a standard curve and compared between sample types.

### 2.3. Virus Isolation

Virus isolation was conducted in porcine primary leukocytes (PPLs), following the NCFAD standard operating protocol for African swine fever virus isolation [24]. Briefly, PPLs were isolated from fresh porcine blood, plated on 24-well plates, 1 mL/well (10^6^ WBC/mL with 0.4% *v*/*v* RBC). Selected SILN 10% homogenates representing each ASF strain included in the study from pigs 1, 2, 3, 20, 21, 23, 26, 27, 28 and 29 were inoculated into PPLs either neat or at 10^−1^ dilution, 0.2 mL/well in duplicate wells. Cultures were incubated at 37 °C in a 5% CO_2_ incubator for 5–7 days and daily observed for the appearance of hemadsorption (HAD). The appearance of HAD was considered a positive indication of an isolate. The limit detection of the NCFAD ASFV isolation protocol is approximately 10^2^ TCID_50_/mL.

### 2.4. Statistical Analysis

The Pearson correlation coefficient between the ASFV genomic copy numbers from the spleen and SILNs was calculated using GraphPad Prism Software, version 9.0.2 (San Diego, CA, USA). The unpaired t-test was used to compare the mean copy numbers between the spleen and SILNs in the highly and moderately virulent ASFV strains. Statistical significance was considered at *p* < 0.05.

## 3. Results and Discussion

In this study, we used whole blood, spleen and SILN samples from pigs infected with both highly virulent (ASFV Ghana 20, ASFV Nigeria RV502, ASFV Vietnam and ASFV Georgia 2007/1) and moderately virulent ASFV Estonia 2014 strains. In all pigs that succumbed to ASF, found dead, or euthanized as they reached a humane endpoint, ASFV genomic material was detected in all three sample types—whole blood (the same day or day before), spleen and SILNs (Table 1)—except for four pigs found dead in the ASF Estonia group that were not bled the previous day.

When the ASFV genome log copy numbers in the spleens were compared with those of the SILNs, there was a positive correlation (*r* > 0.7, *p* < 0.0001) across all ASFV strains (Figure 2). When all moderately virulent and highly virulent ASFV strains were considered collectively, the genome copy numbers in the spleen samples positively correlated with the SILN samples (*r* = 0.77), with high confidence at *p* < 0.0001 (Figure 2a). The comparison among the samples collected from moderately virulent ASFV Estonia 2014-infected pigs indicated the highest level of positive correlation (*r* = 0.85) between spleen and SILNs (Figure 2b). In the samples collected from highly virulent ASFV-infected pigs, a similar positive correlation (*r* = 0.70) was observed (Figure 2c). Overall, regardless of the ASFV strain, SILNs, as a more accessible sample type, showed highly correlated genomic copy numbers to those in the spleen, indicating that they are a suitable sample type for ASFV detection in pigs that succumbed to the disease.

When the mean ASFV genome copy numbers in spleen and SILNs were compared, spleen samples had slightly higher copy numbers than SILNs (Figure 2d–f). However, the difference was not statistically significant when all the samples (both from pigs infected with highly virulent and moderately virulent ASFV strains) were analyzed together (Figure 2d). When the genome copy numbers in samples from pigs infected with moderately virulent ASFV Estonia 2014 strain were analyzed separately (Figure 2e), there was no statistical difference between the mean detections in the two sample types either. However, a wider distribution of genome copy numbers was observed in both spleen and SILN samples in the animals infected with ASFV Estonia 2014 (Figure 2e). In the pigs infected with highly virulent ASFV strains, the mean ASFV genome copy number was the highest in the spleen samples (Figure 2f) at a statistically significant level (*p* = 0.0062). Unlike in the moderately virulent ASFV Estonia 2014-infected group, there was less variability of genome copy numbers observed in both spleen and SILN samples from the pigs infected with the highly virulent strains. Regardless of the genome copy numbers and the statistical significance levels, all animals that tested positive for ASFV genome in the whole blood and spleen samples also tested positive for ASFV genome in their SILNs (100% specificity).

In this study, we show that the pigs infected with moderately virulent ASFV Estonia 2014 showed a wider distribution of genome copies in the spleen and SILN samples. Similar observations were reported by Pikalo et al. in pigs infected with a moderately virulent ASFV Estonia 2014 strain [22]. The study showed high variability of genome detection in lymphoid organs, possibly due to inefficient distribution or rapid clearing of the infection in the lymphoid organs when pigs were infected by less virulent ASFV strains. This could also be due to sampling errors caused by sub-optimal sample collection in the Pikalo et al.’s study from surrounding fatty tissue in which SILNs are embedded.

A selected number of SILNs with high and low ASFV genomic copy numbers representing all the ASFV strains in the study were subjected to virus isolation. The virus was successfully isolated with positive HAD from all SILN samples, including the one with the lowest genomic log copy numbers, i.e., 1.3 in pig 1 (Table 1).

In this study, most of the samples were originated from pigs infected via oro-nasal route (experiments 1, 2, 4 and 5) either experimentally or by contact with infected pigs (study 6). At the NCFAD, in our ASF experimental inoculations, we used a dose of 2 × 10^5^ TCID_50_ for oro-nasal infection to obtain 100% of the animals infected. This dose is higher than the minimum infectious dose of ASFV required to infect pigs, as described by Niederwerder et al. (2019) [28]. Although the potential influence of the higher dose of virus used in this study on the virus load in SILN cannot be ruled out, the route of infection and the different doses used in the current study did not have any influence on the viral load in SILNs or in blood samples collected from pigs that succumbed to the infection.

## 4. Conclusions

In conclusion, we show evidence that SILNs can be used as a reliable sample type for rapid screening of dead pigs for ASF when a complete necropsy is not possible or not desirable. Based on our study, SILN samples from pigs that succumbed to both highly and moderately virulent ASFV strains can be used to detect ASFV genome and for virus isolation. SILN collection from dead pigs requires no highly skilled staff and can be carried out relatively quickly with minimum environmental contamination. The SILN samples tested in the study were collected from the pigs within a few hours of death. However, in the field, some of the carcasses sampled may have initiated to autolyze and this might affect the virus detection, especially virus isolation. ASFV is a highly stable double-stranded DNA virus that persists in decomposing carcasses at room temperature for several months [29,30]. Therefore, we believe that a brief delay (1–2 days) in sample collection is unlikely to affect the detection of ASFV genome by RT-PCR.

The ability to use SILNs for screening dead pigs may enable both ASF-free, as well as ASF-endemic countries, to streamline and expand their passive surveillance for ASF. Even though the data presented in this study are from domestic pigs, the sampling method is equally beneficial and applicable to the ASF surveillance of wild pigs. Testing based on SILNs is not suitable for routine disease investigations in domestic pigs where other infectious agents, such as porcine reproductive and respiratory syndrome virus (PRRSV), porcine circovirus 2 (PCV2), or bacterial infections, are suspected.

Pigs infected with low-virulent strains show a subclinical form of the disease with no mortality [31]; therefore, PCR-based approaches may not be useful for identifying ASF-infected animals.

## Figures and Tables

**Figure 1 viruses-14-00083-f001:**
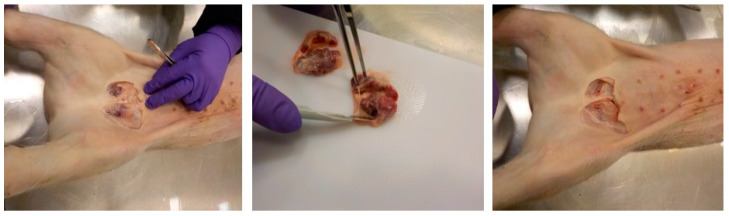
Superficial inguinal lymph nodes are a quickly collectable sample type in pigs with no/minimal bloodshed or full post mortem examination.

**Figure 2 viruses-14-00083-f002:**
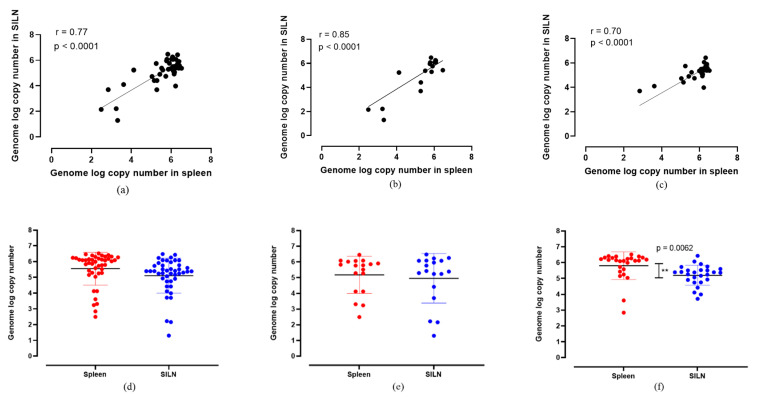
Correlation of ASFV genome log copy numbers detected between spleen and SILNs together with the scatter dot plots indicating the relative distribution of detection levels in all ASF strains collectively (**a**,**d**), ASF Estonia only (**b**,**e**) and highly virulent ASF strains (**c**,**f**). The ASF genome log copy numbers detected in spleens and SILNs correlated positively (*r* = 0.77, *r* = 0.85 and *r* = 0.70) at statistically significant levels (*p* < 0.0001). When the unpaired t-test was performed, there was no significant difference between mean genome log copy numbers, except among the highly virulent ASF strains, where the mean genome log copy numbers were significantly different ** (*p* = 0.0062).

**Table 1 viruses-14-00083-t001:** ASFV real time PCR detections and genomic copy numbers in whole blood, SILNs and spleens of pigs infected with different ASFV strains.

Category	Experiment	ASFV Strain	Pig #	DPI/DPE	Dead or Euthanized	Genome Log Copy # in Blood	Genome Log Copy # in SILNs	Genome Log Copy # in Spleen
Moderately virulent	1	ASFV Estonia 2014	1	10	Euthanized	4.02	1.30	3.31
			2	9	Euthanized	3.50	2.22	3.24
			3	17	Euthanized	3.59	2.16	2.49
	2	ASFV Estonia 2014	4	11	Euthanized	4.27	5.24	4.12
			5	10	Euthanized	5.76	6.08	6.09
			6	9	Euthanasia	5.92	6.09	6.02
			7	8	* Dead	Not collected	6.23	6.09
			8	11	Euthanized	5.70	3.70	5.28
			9	8	Euthanized	5.30	6.08	5.79
			10	11	* Dead	Not collected	5.24	4.12
			11	8	Euthanized	5.46	5.77	5.91
			12	9	Euthanized	6.03	5.43	6.46
			13	8	Euthanized	5.93	5.96	5.77
			14	9	Euthanized	5.92	5.30	5.86
			15	8	Euthanized	5.70	6.26	6.07
			16	10	Euthanized	4.40	4.42	5.29
			17	10	* Dead	Not collected	5.39	5.52
			18	7	* Dead	Not collected	6.48	5.83
Highly virulent	3	ASFV Ghana 20	19	7	Dead	5.13	4.75	5.73
			20	7	Euthanized	5.88	5.73	6.32
	4	ASFV Nigeria RV502	21	7	Euthanized	6.03	6.43	6.32
			22	7	Euthanized	5.58	5.38	6.52
	5	ASFV Vietnam2561	23	9	Dead	5.37	5.51	6.30
			24	12	Euthanized	3.49	4.11	3.61
			25	12	Euthanized	3.15	3.71	2.84
	6	ASFV Georgia 2007/1	26	15	Dead	5.80	5.75	5.25
			27	22	Dead	3.64	5.60	6.41
			28	17	Euthanized	5.64	5.24	5.58
			29	18	Dead	6.09	5.07	6.17
			30	23	Euthanized	5.52	3.99	6.22
			31	14	Dead	5.90	5.36	5.99
			32	16	Euthanized	5.52	4.91	5.43
			33	23	Euthanized	3.87	5.39	6.39
			34	23	Euthanized	5.79	5.26	6.13
			** 35	7	Dead	5.39	6.07	6.24
			36	14	Euthanized	5.54	5.39	6.12
			37	14	Euthanized	5.48	5.51	6.09
			38	13	Euthanized	5.89	4.93	6.15
			39	22	Dead	5.34	5.91	6.39
			40	25	Euthanized	6.09	5.17	6.11
			41	14	Euthanized	5.20	4.74	5.04
			42	13	Dead	5.20	5.41	6.27
			43	15	Euthanized	4.77	4.42	5.15
			44	25	Euthanized	6.00	5.53	6.19

* The pigs that were found dead in the ASFV Estonia 2014 infected group did not have blood samples collected prior to death. In the other groups, the last blood sample collected on the previous day prior to being found dead was considered to be included in the table. ** In the ASF Georgia 2007 infected group, pig 35 was originally infected as the seeder pig. The other pigs mentioned in the table were exposed to the virus shed by the seeder pig; hence, days post exposure (dpe) were considered for them rather than days post-infection (dpi).

## Data Availability

All data related to this study will be made available upon request.

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
