# Peer review of "Superficial Inguinal Lymph Nodes for Screening Dead Pigs for African Swine Fever"

_viruses, 2022, doi:10.3390/v14010083_

Round 1

Reviewer 1 Report

ASFv considers as one of the pathogens of highest concern globally. Consequently is of high importance to be able to perform sampling for surveillance in a way that contamination risk and spreading of virus due to the relevant procedures will be minimized. 

Under this scope, the present paper is considered to be valuable.

Author Response

Reviewer 1 comments

ASFv considers as one of the pathogens of highest concern globally. Consequently is of high importance to be able to perform sampling for surveillance in a way that contamination risk and spreading of virus due to the relevant procedures will be minimized.  Under this scope, the present paper is considered to be valuable.

Response: We thank the reviewer for his comments and supporting this manuscript

Reviewer 2 Report

The study aims to investigate whether superficial inguinal lymph nodes could be used for African swine fever routine surveillance in dead pigs as alternative to OIE recommended samples. Considering the recent spread of ASF in several continents and the importance of surveillance in domestic pigs, a faster and posing less risk of environmental contamination sampling of pigs is of major importance.

While suggested use of superficial inguinal lymph nodes for detection of ASF proved to be useful in experimental infections with highly-virulent and moderately-virulent ASFV strains, its usefulness in field conditions should be still proved.

Minor revisions are suggested:

    In the abstract African swine fever is abbreviated as ASFV (line 16), while later only ASF abbreviation is used.

In the introduction part African swine fever  is abbreviated as ASF, while the abbreviation ASFV is not defined.  

Line 51: references No. 11 and 12 are not really related to OIE recommendations regarding sample types for ASF detection.

    Pictures for Figure 1 should be moved before the explanatory title in line 79. I suggest to include “pig” in the explanatory title of the Figure 1. 

    For Experiment #2 (line 113), Experiment #5 (line 131), and Experiment#6 (lines 139-140) the total volume of infection dose is not indicated. 

    Line 235, number of Figure 1 should be changed to Figure 2.

In the reference list references No. 5, 9 and 18 are missing a journal name.

Author Response

Reviewer 2 comments

The study aims to investigate whether superficial inguinal lymph nodes could be used for African swine fever routine surveillance in dead pigs as alternative to OIE recommended samples. Considering the recent spread of ASF in several continents and the importance of surveillance in domestic pigs, a faster and posing less risk of environmental contamination sampling of pigs is of major importance. While suggested use of superficial inguinal lymph nodes for detection of ASF proved to be useful in experimental infections with highly-virulent and moderately-virulent ASFV strains, its usefulness in field conditions should be still proved.

Minor revisions are suggested:

  1. In the abstract African swine fever is abbreviated as ASFV (line 16), while later only ASF abbreviation is used.

Response: The abbreviation was corrected- Now Line 28

  1. In the introduction part African swine fever is abbreviated as ASF, while the abbreviation ASFV is not defined.

Response:  ASFV is defined  -  Line 42

  1. Line 51: references No. 11 and 12 are not really related to OIE recommendations regarding sample types for ASF detection.

Response: References11 and 12 were removed.

  1. Pictures for Figure 1 should be moved before the explanatory title in line 79. I suggest to include “pig” in the explanatory title of the Figure 1.

Response: Done as requested by the reviewer. Line 82

  1. For Experiment #2 (line 113), Experiment #5 (line 131), and Experiment#6 (lines 139-140) the total volume of infection dose is not indicated.

Response: This information was added. Line 123, Line 143

  1. Line 235, number of Figure 1 should be changed to Figure 2.

Response: This has been corrected

  1. In the reference list references No. 5, 9 and 18 are missing a journal name.

Response: This has been corrected

Reviewer 3 Report

The manuscript by Goonewardene et al. entitled "Superficial inguinal lymph nodes for screening dead pigs for African swine fever" describes the potential use of superficial inguinal lymph nodes (SILN) for ASFV detection in dead pigs. The experiment began with the idea that SILN can be easily collected with no to a little environmental contamination with ASFV. The authors collected SILN, spleen, and blood from dead or euthanized pigs experimentally infected with different ASFV strains, detected viral DNA, and compared the genome copy numbers (GCN) between the spleen and SILN. The results showed that GCN in spleens were highly correlated with GCN in SILN, although the mean ASFV GCN in SILN was significantly lower than that in spleens of pigs infected with highly virulent ASFV strains. It is a well-designed study with the clear objective, the data presented in this paper supported the hypothesis, and the paper was well written. The newly proposed collection method would contribute to ASFV surveillance in the field. It is unlikely that additional experiments are needed, but some concerns should be addressed to provide better context to its findings.

Major concerns

  1. Authors tested clinical samples from pigs that were experimentally infected with different ASFV strains. Genetic or pathogenic features of Estonia 24, Nigeria RV502, Georgia 07 strains were clearly described or cited. However, the information on Ghana 20 and Vietnam 2561 was missing in the manuscript. Please add the information of these strains, such as genotype and isolation year, to the manuscript.
  2. Lines 171-172 and 251-253: Please describe the methods and results of virus isolation in more detail, such as which samples were selected and the titers.
  3. Please describe the limit of detection (LOD) of both PCR and virus isolation assay in the Materials and Methods section.
  4. One of the greatest strengths in this study was that most samples originated from pigs infected with ASFV via oro-nasal route (studies #1, #2, #4, #5) or natural transmission (study #6 except for the seeder pig), rather than intramuscular injection. However, this point might be the weakness because the virus dose for oro-nasal challenge was relatively high, 2 x 105 TCID50/ml. These points need to be discussed in this manuscript.
  5. In modern pig production system, the dead or diseased pig are necropsied for pathological evaluation and sample collection if infectious agents are suspected, because other endemic infectious agents, such as PRRSV, PCV2 and other bacteria, also contribute to pig death or illness. In this respect, the sampling method that was proposed in this study may be more applicable to ASFV surveillance in wild boars because (1) it is required to minimize the environmental contamination during the sample collection, and (2) unskilled workers might be involved in passive surveillance. Although this data were from domestic pigs, the authors can also discuss the possible use for the surveillance in wild boars.
  6. The authors collected and tested the fresh clinical samples under the experimental conditions. However, the situation in the field might not be different from what the authors designed. For instances, the dead pig can be recognized after some period of time, e.g. passive surveillance for dead wild boars. In this case, viral RNA stability in tissues might have an impact on sensitivity of the assay tested. The potential limitations of this study, as mentioned above, need to be discussed to give a clear idea of the practical application of the newly proposed sample collection method to the field.

Minor concerns

  1. Line 79: Figure caption should be placed right below the figure.
  2. Line 235: The figures should be numbered in the order.
  3. Lines 215-218, 222, 225-228, 232, 234: Some line numbers overlap with the figure.

Author Response

Reviewer 3 comments

The manuscript by Goonewardene et al. entitled "Superficial inguinal lymph nodes for screening dead pigs for African swine fever" describes the potential use of superficial inguinal lymph nodes (SILN) for ASFV detection in dead pigs. The experiment began with the idea that SILN can be easily collected with no to a little environmental contamination with ASFV. The authors collected SILN, spleen, and blood from dead or euthanized pigs experimentally infected with different ASFV strains, detected viral DNA, and compared the genome copy numbers (GCN) between the spleen and SILN. The results showed that GCN in spleens were highly correlated with GCN in SILN, although the mean ASFV GCN in SILN was significantly lower than that in spleens of pigs infected with highly virulent ASFV strains. It is a well-designed study with the clear objective, the data presented in this paper supported the hypothesis, and the paper was well written. The newly proposed collection method would contribute to ASFV surveillance in the field. It is unlikely that additional experiments are needed, but some concerns should be addressed to provide better context to its findings.

 Major concerns

  1. Authors tested clinical samples from pigs that were experimentally infected with different ASFV strains. Genetic or pathogenic features of Estonia 24, Nigeria RV502, Georgia 07 strains were clearly described or cited. However, the information on Ghana 20 and Vietnam 2561 was missing in the manuscript. Please add the information of these strains, such as genotype and isolation year, to the manuscript.

Response:  The requested information was added. Lines 130 and 142-143

  1. Lines 171-172 and 251-253: Please describe the methods and results of virus isolation in more detail, such as which samples were selected and the titers.

Response: The method of virus isolation has been elaborated in the manuscript in the Materials and methods section as below. Lines 187 – 195.

“Briefly, PPLs were isolated from fresh porcine blood, plated on 24 well plates, 1 mL/well (106 WBC/mL with 0.4% V/V RBC). Selected SILN 10% homogenates representing each ASF strain included in the study from pigs 1, 2, 3, 20, 21, 23, 26, 27, 28 & 29 were inoculated into PPLs either neat or 10-1 dilution, 0.2mL/ well in duplicate wells. Cultures were incubated at 37 ℃ in a 5% CO2 incubator for 5–7 days and daily observed for the appearance of hemadsorption (HAD). The appearance of HAD was considered a positive indication of an isolate. The limit detection of the NCFAD ASFV isolation protocol is approximately 102 TCID50/ml”

In the Results and Discussion section,  results of virus isolation was described in details as following (Lines 269 – 272).

“A selected number of SILN with high and low ASFV genomic copy numbers representing all the ASFV strains in the study were subjected to virus isolation. Virus was successfully isolated with positive HAD from all SILN samples including the one with the lowest genomic log copy numbers i.e. 1.3 in Pig #1 (Table 1).”

  1. Please describe the limit of detection (LOD) of both PCR and virus isolation assay in the Materials and Methods section.

Response: This information was added to the Material and Methods section. Lines  174-175 and Lines 194-195.

  1. One of the greatest strengths in this study was that most samples originated from pigs infected with ASFV via oro-nasal route (studies #1, #2, #4, #5) or natural transmission (study #6 except for the seeder pig), rather than intramuscular injection. However, this point might be the weakness because the virus dose for oro-nasal challenge was relatively high, 2 x 105 TCID50/ml. These points need to be discussed in this manuscript.

Response: As the reviewer pointed out, one of the greatest strengths in this study is that the most of the samples were originated from pigs infected via oro-nasal route (studies #1, #2, #4, #5)  either experimentally or by contact with infected pigs (study #6). At the NCFAD, we use a dose of 2X105 TCID50 for oro-nasal infection in our ASF experimental inoculations to ensure 100%  of the animals inoculated get infected. We agree with the reviewer that this dose is higher than the minimum infectious dose of ASFV required to infect pigs, i.e. 100  TCID50 in liquid and 104 TCID50 in feed as described by Niederwerder et al. 2019 (doi:10.3201/eid2505.181495). Possible influence of the higher dose of virus used in this study on the virus load in SILN can’t be ruled out. However, the  route of infection and the different doses used in the current study did not have any influence on viral load  in  SILN or in blood collected from pigs succumbed to the infection. 

We have discussed this in the manuscript – Lines: 273-281.

  1. In modern pig production system, the dead or diseased pig are necropsied for pathological evaluation and sample collection if infectious agents are suspected, because other endemic infectious agents, such as PRRSV, PCV2 and other bacteria, also contribute to pig death or illness. In this respect, the sampling method that was proposed in this study may be more applicable to ASFV surveillance in wild boars because (1) it is required to minimize the environmental contamination during the sample collection, and (2) unskilled workers might be involved in passive surveillance. Although this data were from domestic pigs, the authors can also discuss the possible use for the surveillance in wild boars.

Response:  We agree with the reviewer that in routine disease investigations, dead pigs in modern pig production systems should also be screened  for pathogens such as PRRSV, PCV2, bacterial infections etc. Here we propose that SILN testing can be used on targeted ASF surveillance exercises, that require testing large numbers of pigs at national and or regional level.  Similarly, as the reviewer suggested, SILN testing can be used in ASFV surveillance of wild pigs  and it will minimize the environmental contamination during sample collection and requirement  of highly skilled workers. Therefore, even though  this data is from domestic pigs, the sampling method is equally beneficial and applicable to the ASF surveillance in wild pigs.

We have discussed this in the manuscript-  Lines 298-303

  1. The authors collected and tested the fresh clinical samples under the experimental conditions. However, the situation in the field might be different from what the authors designed. For instances, the dead pig can be recognized after some period of time, e.g. passive surveillance for dead wild boars. In this case, viral RNA stability in tissues might have an impact on sensitivity of the assay tested. The potential limitations of this study, as mentioned above, need to be discussed to give a clear idea of the practical application of the newly proposed sample collection method to the field.

Response: We agree with the review that the SILN samples tested in the study were collected relatively fresh (we had some animals that died overnight). However in the field, especially with wild pig surveillance some of the carcass may have initiated to autolyze and this may affect the virus detection in these animals irrespective of the tissues type. However  ASFV is a highly resistant double stranded DNA virus that is fairly stable in a range of temperatures and it persists in decomposing carcasses at room temperature for several months.

We have discussed this in the manuscript and have two new references Fischer et al. 2020 and  Mazur-Panasiuk et al., 2019) -  Lines 290-296

Minor concerns:

  1. Line 79: Figure caption should be placed right below the figure.

Response: Figure caption has been placed below the figure

  1. Line 235: The figures should be numbered in the order.

Response: This correction has been made

  1. Lines 215-218, 222, 225-228, 232, 234: Some line numbers overlap with the figure.

Response: This issue has been resolved

Round 2

Reviewer 3 Report

The manuscript has been improved and is suitable for publication.